# Increasing temperature can modify the effect of straw mulching on soil C fractions, soil respiration, and microbial community composition

Xin Fu[1,2], Jun Wang[1,3]*, Mengyi Xie[1,3], Fazhu Zhao[1,3], Russell Doughty[4]

1 State Key Laboratory of Soil Erosion and Dryland Farming on the Loess Plateau, Institute of Soil and Water Conservation, Chinese Academy of Sciences and Ministry of Water Resources, Yangling, Shaanxi, China, 2 College of Land and Resources, Hebei Agricultural University, Baoding, China, 3 Shaanxi Key Laboratory of Earth Surface System and Environmental Carrying Capacity, College of Urban and Environmental Science, Northwest University, Xi'an, China, 4 Division of Geological and Planetary Sciences, California Institute of Technology, Pasadena, California, United States of America

* wangj@nwu.edu.cn

**Data Availability Statement:** The raw sequences for all samples were sent to the Sequence Read Archive (SRA) database of the National Center for Biotechnology Information (NCBI) with the

## Abstract

Straw mulching has been widely adopted in dryland cropping but its effect on soil respiration and microbial communities under warming are not well understood. Soil samples were collected from a corn field with straw mulching (SM) for nine years and without straw mulching (CK), and incubated at 15°C, 25°C, and 35°C for 60 days. Soil respiration, C fractions and bacterial and fungal community structure were measured SM had greater soil organic carbon and potential C mineralization and a similar microbial biomass carbon throughout the incubation when compared with CK. Soil respiration increased with increasing temperature and its temperature sensitivity ($Q_{10}$) was lower with SM than CK. Similar microbial community composition was found in the soils with SM and CK before incubation. However, SM had a greater bacterial richness and the relative abundances of *Proteobacteria*, *Acidobacteria*, *Nitrospirae*, *Planctomycetes*, *Bacteroidetes*, and *Basidiomycota*, but lower relative abundances of *Actinobacteria*, *Chloroflexi*, *and Ascomycota* than CK after incubation. Bacterial richness and diversity were greater at 15°C and 25°C than 35°C, but there was no difference in fungal richness and diversity among the incubation temperatures. As temperature increased, the relative abundances of *Chloroflexi*, *Acidobacteria*, and *Bacteroidetes* decreased, but *Gemmatimonadetes* and *Ascomycota* increased, and were significantly correlated with soil C fractions and respiration. These findings indicated that the effect of straw mulching on soil C cycling and microbial community structure can be highly modified by increasing temperature.

## Introduction

Soil contains around 1500 Pg of organic carbon (C) and plays a major role in the carbon cycling in terrestrial ecosystems [1]. A small variation in soil C sequestration can lead to a significant change in atmospheric $CO_2$ concentration [2]. Increased soil C storage has been

accession numbers SRP260944 for bacteria and SRP261054 for fungi. All other relevant data are within the manuscript and its Supporting Information files.

**Funding:** The study was supported by the Chinese Academy of Sciences 505 "Light of West China" Program for Introduced Talent in the West, the National Natural Science Foundation of China (Grant No. 507 31570440, 31270484), and the Key International Scientific and Technological Cooperation and Exchange Project of Shaanxi Province, China (grant number 2020KWZ-010)

**Competing interests:** The authors have declared that no competing interests exist.

suggested as a way to mitigate greenhouse gas emissions [3]. In the recent decades, straw mulching has widely been adopted to conserve soil water, regulate soil temperature and increase crop yield in dryland cropping systems. The application of straw mulch also has been proposed as a method to store organic carbon long term [4, 5]. In an 8-yr study in the Loess Plateau of China, Wang et al. [5] reported that soil organic C (SOC) stock was 7–35% greater with straw mulching than without. Generally, SOC would change slowly with management practices due to its large pool sizes and inherent spatial variability [4]. Soil labile C fractions, such as microbial biomass carbon (MBC) and potential C mineralization (PCM) would response more rapidly to environmental change than SOC [6].

Straw mulching increased substrate availability for soil microbials due to additional input [7]. Similarly, soil hydrothermal conditions can be significantly changed with straw mulching as related to no mulching [8, 9]. Such changes in soils due to straw mulching would affect soil respiration (SR) and microbial activities. Several studies reported that straw mulching could increase the SR rate due to higher availability of C substrates [10]. However, SR rate was controlled by many factors, such as soil temperature, moisture, and nitrogen levels. The studies about the effect of straw mulching on SR rate had different results [11]. Soil microorganisms are the main decomposer groups involved in the soil C cycling [3]. Most previous studies showed that straw mulching can increase the activity and population of soil bacteria and fungi, due to higher soil C substrate quality and quantity [12, 13]. But contrasting results were also reported, that soil microbial diversity was significantly lower with straw mulching than without [14].

In the context of climate change, the responses of SOC, SR and microbial activity to warming have gained more attentions recently. Increasing temperature would stimulate soil microbial metabolisms [15], accelerate SOC decomposition [16] and increase C efflux through SR [17]. Since the temperature sensitivity of SR ($Q_{10}$) varied with the substrate availability [18], how the comprehensive responses of C soil fractions, SR and microbial communities to straw mulching would vary with different temperature change is now well reported. Here, we carried out an incubation study, aimed to (1) determine changes in carbon fractions and SR rates to different temperatures in soils with and without 9-yr straw mulching, (2) quantify the effect of straw mulching on soil temperature sensitivity; and (3) explore the relationships among soil C fractions, SR rates and the soil microbial community. We hypothesized that: (1) straw mulching would increase SR rates by increasing the input of organic matter in the soil compared with no mulching; and (2) straw mulching could change the temperature sensitivity of the agro-ecosystem due to the regulation of soil C fractions and soil microbial community.

## Materials and methods

### Experimental sites and soil sampling

A mulching experiment was started in 2009 at the Changwu Agro-Ecological Station in the Loess Plateau (107˚ 44.70' E, 35˚ 12.79' N) of China. The site has a monsoon climate with a mean annual temperature of 9.1 ˚C and an annual precipitation of 580 mm. The mean frost-free period is 194 days and the open-pan evaporation is 1440 mm. The soil was a Heilutu silt loam (Calcarid Regosol according to the FAO classification system), with 35 g kg$^{-1}$ sand, 656 g kg$^{-1}$ silt, and 309 g kg$^{-1}$ clay, 1.30 Mg m$^{-3}$ bulk density, 8.3 pH, 9.10 g kg$^{-1}$ SOC, and 1.10 g kg$^{-1}$ soil total nitrogen at 0–20 cm depth at the initiation of the experiment.

The field experiment design has been described in detail by Wang et al. [5]. Briefly, field plots with straw mulch (SM) and no mulch (CK) were arranged in a randomized complete block design with three replications. Each plot size was 6.7 m wide by 10.0 m long. Plots and blocks were separated by 0.5 and 1.0 m strips, respectively. SM included a surface of whole

corn straw at 9000 kg ha$^{-1}$. Corn straw had a C/N ratio of 40.1 and the contents of cellulose, hemicellulose and lignin were 32%, 28% and 15.5%, respectively. Corn was planted in mid-April and harvested in late September for each year. Corn was planted by hand under conventional tillage which consisted of hand tractor-drawn plows to a depth of 10 cm at planting. Nitrogen fertilizer as urea (46% N) at 120 kg N ha$^{-1}$ and phosphate fertilizer as calcium superphosphate (20% P) at 60 kg P ha$^{-1}$ were broadcast and then incorporated to a depth of 20 cm using a rotary tiller before sowing. Potassic fertilizer was not applied because of high soil potassium content (about 130 mg kg$^{-1}$ at 0–20 cm soil depth). Corn was planted at 0.04 million plants ha$^{-1}$ with 60 cm row spacing for all treatments. No irrigation was applied. After crop harvest, left-over straw mulch was removed from the soil surface.

Fresh soil samples were collected from after corn harvest in October 2017. Soil samples (about 10 kg) were collected with a spade from the surface layer (0–20 cm) from five places within a plot. Then we composited five samples to one sample, and placed them in plastic boxes. We tried to avoid damaging the soil structure during the collection process. After removing roots and rocks, the samples were sieved through a 2 mm mesh immediately and then kept at 4 ˚C for incubation.

## Design of the incubation study

Soil samples of 500 g were placed in a jar and incubated at 15 ˚C, 25 ˚C, and 35 ˚C, each with three replicates, for a total of 18 samples. This range of temperatures was selected according to the range of air temperatures that occurred during the crop growing season. The jars were placed in incubators and soil samples were adjusted to 60% water filled pore space using weighing method throughout the incubation [19]. We maintained constant soil moisture by weighing each sample once a week and adjusting the water content to the target mass. Air samples from the headspace of the sealed mason jars were drawn through septa, transferred to evacuated vials, and $CO_2$ concentrations were measured using a Li-COR LI-840A infrared gas analyzer [20]. The processed soils were then subjected to incubation and moisture for 60 days. Carbon dioxide in the headspace of each jar was measured every day at the outset of the incubation.

About 50 g of soil samples were removed from the incubator at 0 and 60 days after incubation (DAI), then air-dried at room temperature, and subjected to measure soil organic carbon (SOC), potential C mineralization (PCM), and microbial biomass C (MBC). Furthermore, fresh soil samples were collected before and at the end (day 60) of the short-term incubation to measure the structural composition of both the bacterial and fungal communities. The fresh soil samples for microbial analysis were stored at -80 ˚C for DNA extraction.

## Soil analysis

SOC concentration (before and during incubation) was measured using a high induction furnace C and N analyzer (Euro Vector EA3000, Manzoni, Italy) after pretreating the soil with 1 mole L$^{-1}$ HCl to remove inorganic C. The PCM and MBC concentrations were determined using the fumigation-incubation method reported by Wang et al. [5]. Briefly, 10 g air-dried soil was moistened with water at 50% field capacity and placed in a 1 L jar containing beakers with NaOH to trap evolved $CO_2$, and incubated in the jar at 21 ˚C for 10 d. PCM concentration was determined by measuring $CO_2$ absorbed in NaOH. The moist soil used for determining PCM was subsequently used for determining MBC by the modified fumigation-incubation method for air-dried soils. The moist soil was fumigated with ethanol-free chloroform for 24 h and placed in a 1 L jar containing beakers with NaOH. Fumigated moist soil was incubated for 10 d, and the $CO_2$ absorbed in NaOH was back-titrated with HCl.

Soil DNA was extracted from 0.5 g of freeze-dried soil using Fast DNA SPIN extraction kits (MP Biomedicals, Santa Ana, CA, USA). The extraction method was same as Ren et al. [2]. The universal Eubacterial primers 338F (5′-ACTCCTACGGGAGGCAGCA) and 806R (5′-GGACTACHVGGGTWTCTAAT-3′) were used for amplifying the 16S rRNA V3-V4 fragment. The universal eukaryotic primers ITS5F (5′-GGAAGTAAAAGTCGTAACAAGG) and ITS1R (GCTGCGTTCTTCATCGATGC) were used for amplifying the ITS-1 region. The 7-bp barcodes were incorporated into the primers for multiplex sequencing. The solution for bacterial amplification contained 0.4 µl of each primers, 0.4 µl FastPfu polymerase, and 10-ng template DNA. Thermal cycling consisted of initial denaturation at 98 ˚C for 2 min, 25 cycles (98 ˚C for 15 s, 55 ˚C for 30 s, and 72 ˚C for 30 s), followed by a final extension at 72 ˚C for 5 min. The PCR reaction of ITS rRNA was carried out in a 25-µl mixture which contained 0.5µl of each primer at 30 µmol l$^{-1}$, 10-ng template DNA, and 22.5 µl of Platinum PCR SuperMix (Invitrogen, Shanghai, China). PCR reactions for the fungal ITS region were 95 ˚C for 2 min, 30 cycles (95 ˚C for 30 s, 55 ˚C for 30 s, 72 ˚C for 30 s), and followed by a final extension at 72 ˚C for 5 minutes.

Each sample was amplified for three times and the relative amplicons were mixed to provide a final PCR product. Each sample was amplified three times, and then the relative amplicons were mixed to obtain one final PCR product. In order to improve the quality and concentration of the PCR product, each mixed gene was subjected to electrophoresis in 2% agarose gels. PCR products were further excised using an AxyPrep DNA Gel Extraction Kit (Axygen Biosciences, USA), and the relative DNA was solubilized with ddH$_2$O. Finally, PCR products were pooled in a single tube and then sequenced using Illumina's MiSeq platform at the Personal Biotechnology Co., Ltd (Shanghai, China). Raw sequences were processed using the Quantitative Insights into Microbial Ecology (QIME, v1.8.0) pipeline as described by Caporaso et al. [21]. Finally, the raw sequences for all samples was sent to the Sequence Read Archive (SRA) database of the National Center for Biotechnology Information (NCBI, Bethesda, MD, USA) under the accession numbers SRP260944 and SRP261054 for bacteria and fungi, respectively.

## Data analysis

Data for soil C fractions, cumulative respiration, and soil microbial communities were analyzed using a two-way analysis of variance (ANOVA) (SPSS Statistics). The mulching treatment and incubation temperature were considered fixed effects and replication as the random effect for data analysis. Means were separated using Duncan's multiple range test when treatments and interactions were significant. Statistical significance was observed at $P \leq 0.05$. Community taxonomic alpha richness and diversity (Chao1 and Shannon index) were calculated by the mothur software (Version v.1.30.1) [22]. Principal component analysis (PCA), using weighted UniFrac distances,was used to explore the differences in soil bacterial and fungal community structure across the mulching methods and temperatures. Redundancy analysis (RDA) was performed to gain insights into the relationship between the composition of soil bacterial and fungal communities under straw mulching, no mulching, and incubation temperature using CANOCO software [23].

The temperature sensitivity ($Q_{10}$) of soil respiration were determined using the formulas as follows [24]:

$$R_s = ae^{bT}$$

$$Q_{10} = e^{10b}$$

Where $R_s$ is the soil respiration rate (mol m$^{-2}$ d$^{-1}$), $T$ is the temperature (˚C), and $a$ and $b$ are the coefficients. $Q_{10}$ is estimated based on soil respiration rates under temperature increase from 15˚C to 25 ˚C and 25 ˚C to 35 ˚C, separately. The one-way ANOVA and Pearson correlation were analyzed using the SPSS software.

The microbial metabolic quotient ($q$CO$_2$) was determined using the methods by Wardle and Ghani [25].

# Results

## Soil carbon fractions

At the beginning of incubation, SOC and PCM contents were greater with SM than CK (Table 1), and no difference in MBC was found between SM and CK. At the end of the short-term incubation, SOC and PCM contents were greater in SM than CK when averaged across incubation temperatures, and no difference in MBC was found between SM and CK (Table 2 and S1 Table). On average, the contents of SOC, PCM, and MBC were greater with 15 ˚C and 25 ˚C than 35 ˚C. Furthermore, the PCM content was greater after incubation than before incubation in both SM and CK, while the MBC content decreased significantly after short-term incubation.

**Table 1. Soil carbon fractions before incubation.**

| Mulching [a] | SOC (g/kg) | PCM (mg/kg) | MBC (mg/kg) |
|---|---|---|---|
| CK | 8.86b[b] | 207b | 441a |
| SM | 9.55a | 321a | 402a |

[a] CK: no mulching; SM: straw mulching.

[b] Different lowercase letters indicate significant difference between straw mulching and no mulching.

**Table 2. Soil carbon fractions after incubation.**

| Mulching [a] | Incubation temperature (˚C) | SOC (g/kg) | PCM (mg/kg) | MBC (mg/kg) |
|---|---|---|---|---|
| CK | | 8.20b | 326b | 379a |
| SM | | 8.81a | 374a | 360a |
| | 15 | 8.61a | 360a | 406a |
| | 25 | 8.51b | 363a | 359ab |
| | 35 | 8.38c | 327b | 344b |
| **Significance** | | | | |
| Mulch (M) | | *** | ** | NS |
| Temperature (T) | | *** | * | * |
| M×T | | NS | NS | NS |

[a] CK: no mulching; SM: straw mulching.

[b] Different lowercase letters indicate significant difference between straw mulching and no mulching or among incubation temperatures.

*** significant at $P \leq 0.001$ levels;

** significant at $P \leq 0.01$ levels;

* significant at $P \leq 0.05$ levels; NS, no difference.

SOC: soil organic carbon; PCM: potential C mineralization; MBC: microbial biomass carbon.

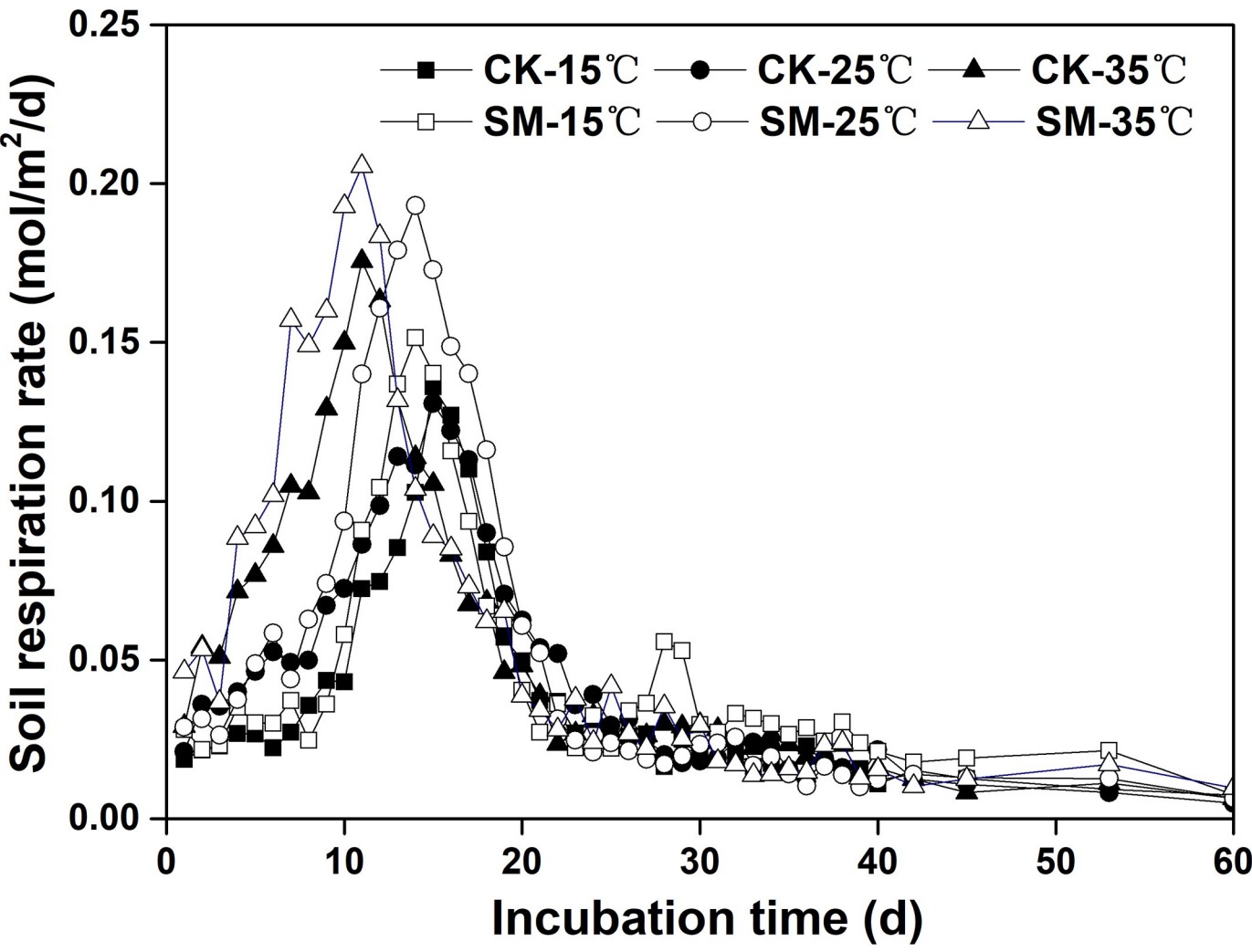

**Fig 1. Soil respiration rate under different mulching methods and incubation temperatures.** CK: no mulching; SM: straw mulching.

## Soil respiration, temperature sensitivity ($Q_{10}$) and metabolic quotient ($q$CO₂)

The rate of SR increased rapidly during early incubation and peaked at 10 DAI with 35 ˚C and at 14–15 DAI with 15 ˚C and 25 ˚C and then declined (Fig 1). Averaged across incubation temperature, the mean respiration rate during 0–15 DAI was greater in SM than CK. At the end of incubation, the respiration rates were only about 50% of the initial values and the 60-d cumulative $CO_2$ emission was significantly greater in SM than CK (Fig 2). The cumulative respiration was 1.78, 2.11, and 2.41 mol m$^{-2}$ at 15 ˚C, 25 ˚C, and 35 ˚C ($P$<0.05), respectively. Strong correlations were found between the cumulative respiration and incubation temperature (r$^2$, 0.967 to 0.998; $P$<0.001), with a $Q_{10}$ of 1.18 and 1.12 for CK and SM, respectively (Table 3). The $q$CO₂ under both CK and SM appeared to increase with increasing incubation temperatures (Fig 3). For all incubation temperatures, SM had higher $q$CO₂ than CK.

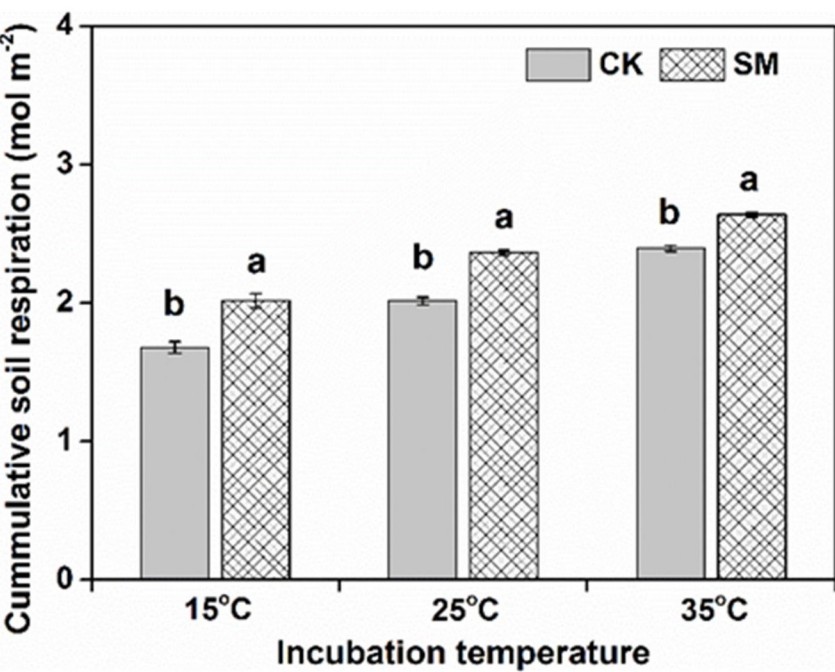

**Fig 2. Cumulative soil respiration over the 60-day incubation under straw mulching and no mulching at different incubation temperatures.** a CK: no mulching; SM: straw mulching. b Different lowercase letters indicate significant difference between straw mulching and no mulching.

## Soil microbial diversity and community structure

Observed species and bacterial alpha-diversity indices based on the Chao 1 richness and Shannon's diversity indices were not significantly different with and without straw mulch before incubation (Table 4). SM had greater bacterial richness compared to CK when averaged across incubation temperatures. The bacterial richness and diversity were greater in 15 ˚C and 25 ˚C than 35 ˚C (Table 5).

Principal coordinates analysis showed no difference in the OTU composition between CK and SM before incubation (S1 Fig). However, both bacteria and fungi varied with mulching treatments and incubation temperature after incubation (Fig 4). About 55.9–61.3% and 66.2–69.7% of the variance was explained by the first two axes for bacteria and fungi after incubation, respectively. According to PCA analysis, bacterial and fungal communities were significantly different from each other with and without straw mulch and for the incubation temperatures (Figs 4 and 5).

**Table 3. Regression equations between cumulative Soil Respiration (R$_s$) and incubation Temperature (T) under no mulching and straw mulching.**

| Mulching | Equation | Q$_{10}$ | R$^2$ | Significance |
|---|---|---|---|---|
| CK[a] | R$_s$ = 1.44e$^{0.0163T}$ | 1.18 | 0.998 | <0.001 |
| SM | R$_s$ = 1.94e$^{0.0115T}$ | 1.12 | 0.967 | <0.001 |

[a] CK: no mulching; SM: straw mulching.

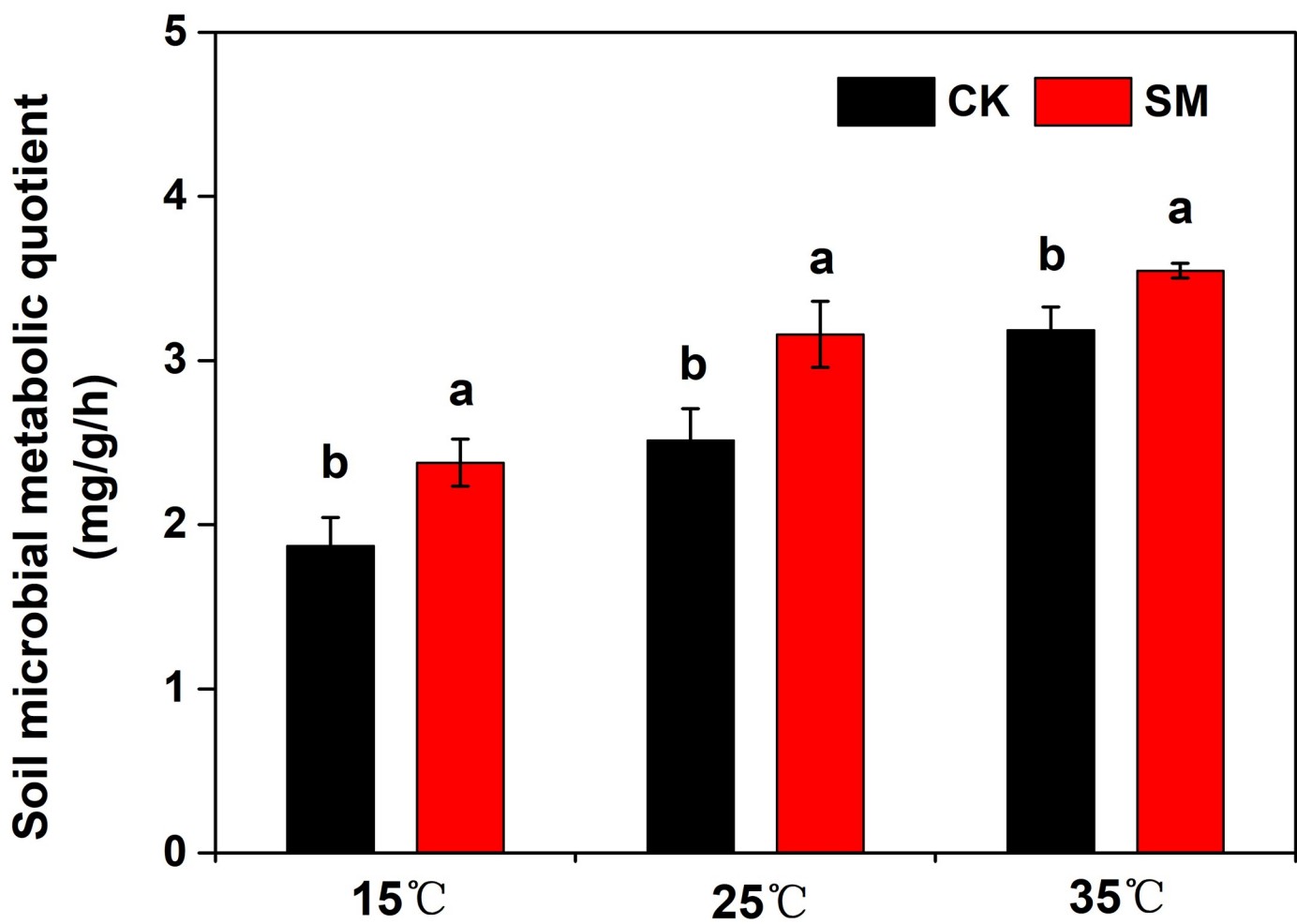

**Fig 3. Soil microbial metabolic quotient ($q$CO$_2$) under different mulching treatments and incubation temperatures.** CK: no mulching; SM: straw mulching.

## Bacterial and fungal community compositions

For all soil samples (before and after incubation), the dominant bacterial phyla ($\geq$1%) consisted of *Proteobacteria* (29.5%), *Actinobacteria* (19.2%), *Chioroflexi* (14.8%), *Acidobacteria* (14.4%), *Gemmatimonadetes* (8.49%), *Nitrospirae* (4.38%), *Planctomycetes* (4.18%), and *Bacteroidetes* (1.52%) (Fig 6). Before incubation, no significant difference in bacterial or fungal phyla was found between CK and SM. However, after incubation, the relative abundances of phyla *Proteobacteria*, *Acidobacteria*, *Nitrospirae*, *Planctomycetes*, and *Bacteroidetes* were greater than that of phyla *Actinobacteria*, and *Chloroflexi* was lower in SM than CK (Fig 6 and S2 Table). For *Proteobacteria*, the dominant classes were *Gammaproteobacteria*, *Alphaproteobacteria*, *Betaproteobacteria*, and *Deltaproteobacteria*, and only *Deltaproteobacteria* increased with straw mulching at all incubation temperatures (S3 Table). Within *Deltaproteobacteria*, the order *Desulfurellales* increased with straw mulch. However, the class *Actinobacteria*, *Thermoleophilia*, and *MB-A2-108*, a branch of *Actinobacteria*, declined greatly with straw mulch. For *Chloroflexi*, class *KD4-96*, *Chloroflexia*, *Thermomicrobia*, and *Gitt-GS-136* were lower in SM than CK.

**Table 4. Soil bacterial and fungal diversity (Shannon index) and richness (Chao1 index) under different mulching methods before incubation.**

| Mulching [a] | Bacteria | | Fungi | |
|---|---|---|---|---|
| | Chao1 | Shannon | Chao1 | Shannon |
| CK | 3628a[b] | 7.58a | 891a | 4.47a |
| SM | 3678a | 7.58a | 970a | 4.61a |

[a] CK: no mulching; SM: straw mulching.

[b] Different lowercase letters indicate significant difference between straw mulching and no mulching.

The microbial relative abundance of phyla and class varied with different incubation temperatures (Fig 6 and S2 and S3 Tables). The relative abundances of phyla *Chloroflexi*, *Acidobacteria*, and *Bacteroidetes* decreased with incubation temperature, while the relative abundance of the phylum *Gemmatimonadetes* was greater at 35 ˚C than at 15 ˚C and 25 ˚C. For bacteria, the relative abundance class of *Alphaproteobacteria*, a branch of *Proteobacteria*, was greater at 15 ˚C than at 25 ˚C and 35 ˚C under CK, but was greater at 15 ˚C and 25 ˚C than at 35 ˚C for SM. The relative abundance class *Betaproteobacteria* was greater at 35 ˚C than at 15 ˚C and 25 ˚C under SM, and had no difference for CK. For *Actinobacteria*, the relative abundance of *Acidimicrobiia* decreased with increasing incubation temperatures under SM. The relative abundance of phyla *Chloroflexi* in CK and *Acidobacteria* in SM were greater at 15 ˚C and 25 ˚C than at 35 ˚C.

The dominant fungal phyla consisted of *Ascomycota* (73.8%), *Basidiomycota* (9.71%), and *Zygomycota* (5.49%) (Fig 6). For fungi, when averaged across incubation temperature, the relative abundance of phylum *Ascomycota* was lower, and *Basidiomycota* was greater in SM than CK (Fig 6, S4 and S5 Tables). For the phylum *Ascomycota*, the class *Sordariomycetes* was the most dominant class and was similar in CK and SM. The relative abundance of the phylum *Basidiomycota* was greater in SM than CK at 15 ˚C and 25 ˚C. The relative abundance of the fungi *Ascomycota* increased with increasing incubation temperatures. No significant difference in other dominant fungal phyla was found among the different incubation temperatures.

**Table 5. Soil bacterial and fungal diversity (Shannon index) and richness (Chao1 index) under different mulching methods after incubation.**

| Mulching [a] | Incubation temperature (˚C) | Bacteria | | Fungi | |
|---|---|---|---|---|---|
| | | Chao1 | Shannon | Chao1 | Shannon |
| CK | | 4810b | 7.66a | 1009a | 4.59a |
| SM | | 5264a | 7.68a | 1048a | 4.77a |
| | 15 | 5539a | 7.78a | 900a | 4.75a |
| | 25 | 4926ab | 7.72a | 1073a | 4.78a |
| | 35 | 4647b | 7.51b | 1112a | 4.51a |
| Significance | | | | | |
| Treatment (T) | | * | NS | NS | NS |
| Temperature (ST) | | * | ** | NS | NS |
| T×ST | | NS | NS | NS | NS |

[a] CK: no mulching; SM: straw mulching.

[b] Different lowercase letters indicate significant difference between straw mulching and no mulching or among incubation temperatures.

** Significant at $P \leq 0.01$ levels;

* significant at $P \leq 0.05$ levels; NS, no difference.

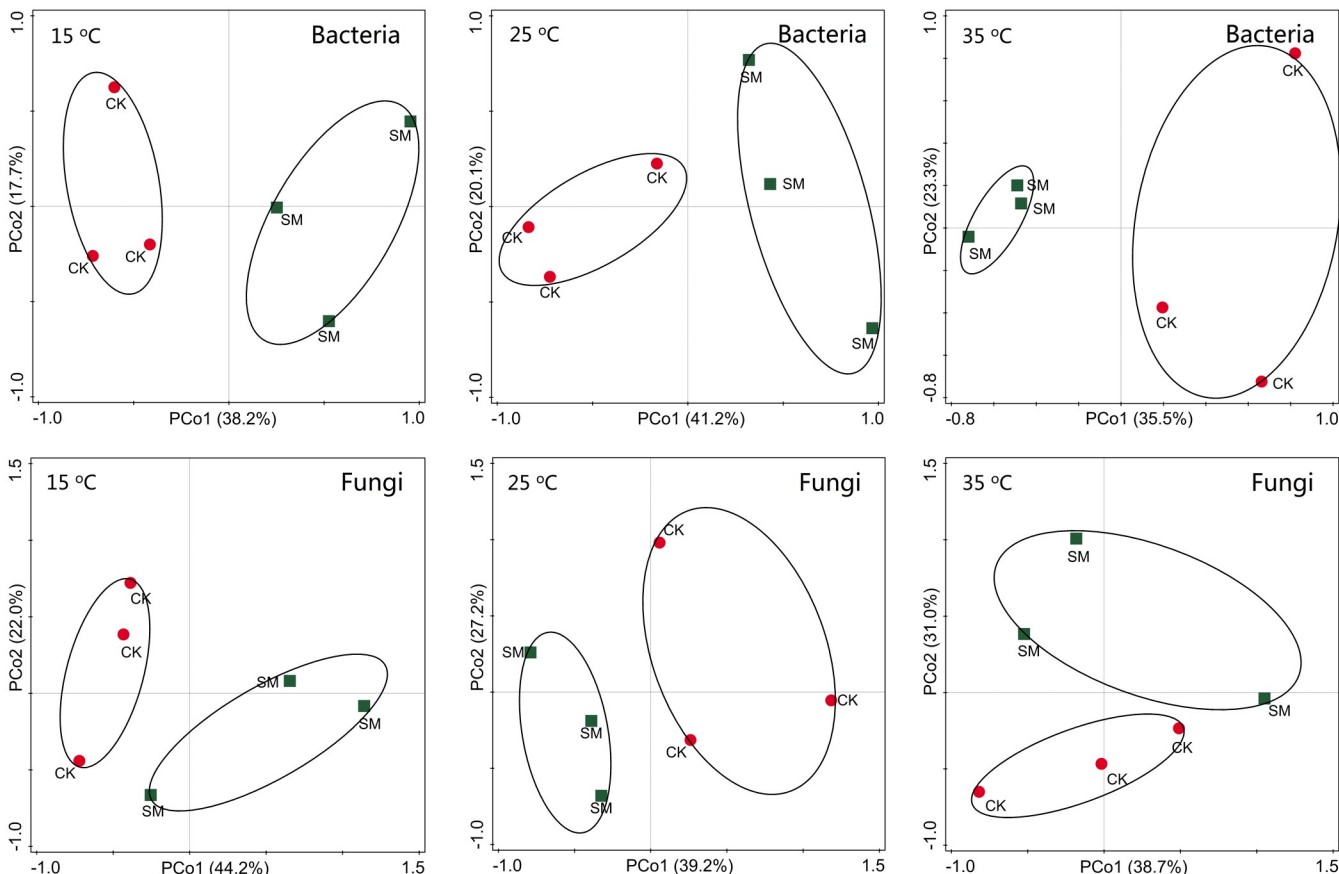

**Fig 4. Principal Coordinates Analysis (PCoA) of soil microbial community composition under straw mulching and no mulching after incubation based on Bray-Curtis distances.** CK: no mulching; SM: straw mulching.

Within the phylum *Ascomycota*, the relative abundances of the classes *Dothideomycetes* and *Lecanoromycetes* increased as incubation temperatures increased. The relative abundance of *Agaricomycetes*, a branch of the phylum *Basidiomycota*, was greater at 15 ˚C and 25 ˚C than at 35 ˚C under SM, and had no difference under CK.

## Relationships among respiration, soil C fraction, and microbial community composition

Soil respiration was highly correlated to SOC for all incubation temperatures ($P<0.01$), and was correlated with PCM at 15 ˚C and 25 ˚C ($P<0.05$) (S6 Table). Redundancy analysis showed strong relationships among soil C fractions, soil respiration, and microbial compositions (Fig 7). SOC and PCM were positively correlated with the relative abundances of *Acidobacteria*, *Bacteroidetes*, *Nitrospirae*, and *Planctomycetes*, and SOC was negatively correlated with *Actinobacteria*. The MBC was positively correlated to the relative abundance of *Chloroflexi*. SR was positively related to relative abundance of *Proteobacteria*, and negatively to the relative abundance of *Chloroflexi*. For the dominant fungal phyla, SOC and PCM were positively related to the relative abundances of *Basidiomycota* and *Zygomycota*, and negatively to that of *Ascomycota*. Furthermore, the correlations between SR, C fractions, and microbial community structure varied with and without straw mulching at both the phylum and class

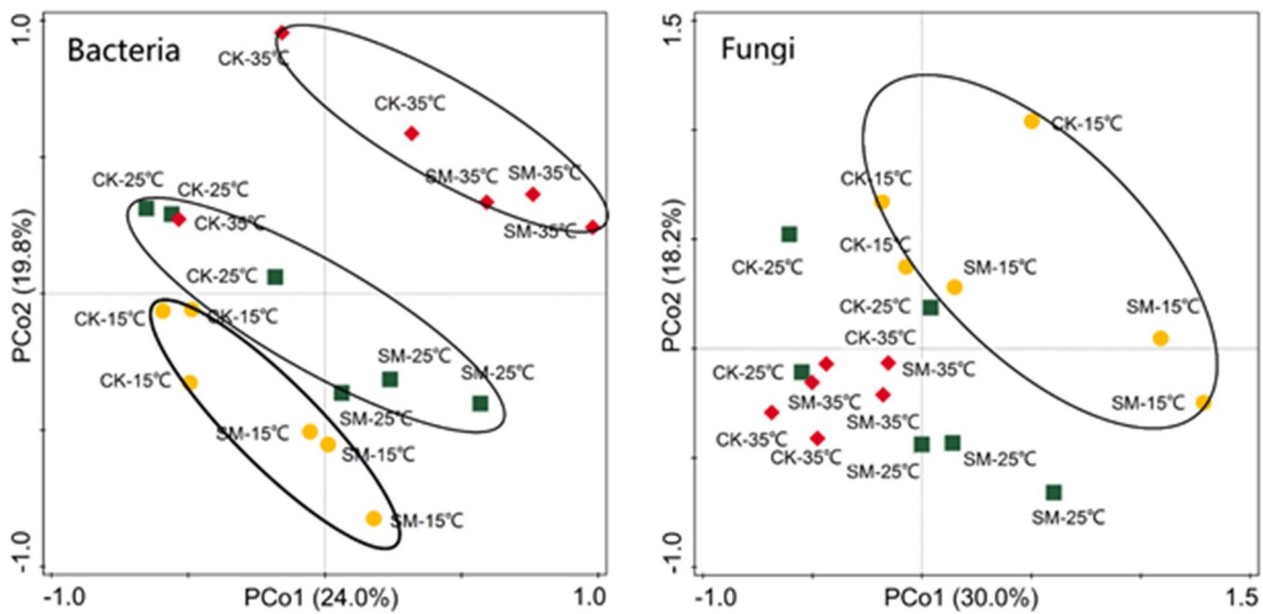

**Fig 5. Principal Coordinates Analysis (PCoA) of soil bacterial community composition at different incubation temperatures based on Bray-Curtis distances.** CK: no mulching; SM: straw mulching.

levels (S7 and S8 Tables). At the phylum level, SOC fractions and/or SR were significantly correlated to the abundance of *Chloroflexi* in CK, and with abundances of *Proteobacteria*, *Acidobacteria*, *Bacteroidetes*, and *Ascomycota* in SM.

## Discussion

The significant decrease in SOC after incubation (Tables 1 and 2) was in accordance with Follett et al. [26]. The amount of SOC present at the beginning of the incubation was indicative of a larger pool of the less resistant fractions that were available to be broken down and recycled, thus resulting in lower percentages of the original SOC remaining after 60 days of incubation [27]. During incubation, the higher SOC concentrations in SM compared to CK could be due to the increased C input from straw mulch (Table 2), which was confirmed by a previous study [5]. Throughout incubation, the reduction of SOC was greater with higher temperature than lower temperatures, probably due to higher soil microbial activity at 35 ˚C. Fissore et al. [27] also showed that cool temperatures reduced the rate of decomposition, resulting in high SOC accumulation. Furthermore, Joergensen et al. [28] found that increasing the temperature from 15 ˚C to 25 ˚C, and further to 35 ˚C, can double and triple the rate of mineralizing soil organic C. Allison et al. [29] suggested that the SOC response to temperature is dependent on how microbial physiology and communities adapt to the new environments, which may lead to an upward adjustment of C utilization and accelerated SOC loss. PCM concentration increased during incubation for all treatments, probably due to the lower microbial activity caused by cooler temperatures before incubation, and microbial activity recovered after incubation when temperature and moisture became favorable. Before and after incubation, PCM was greater with straw mulching than without mulching, which was consistent with the findings reported by Wang et al. [5].

Although soil microbial activity was low at the beginning of incubation (Table 1), it proliferated quickly with the incubation temperature at 15 ˚C, 25 ˚C, and 35 ˚C (Fig 1). Soil

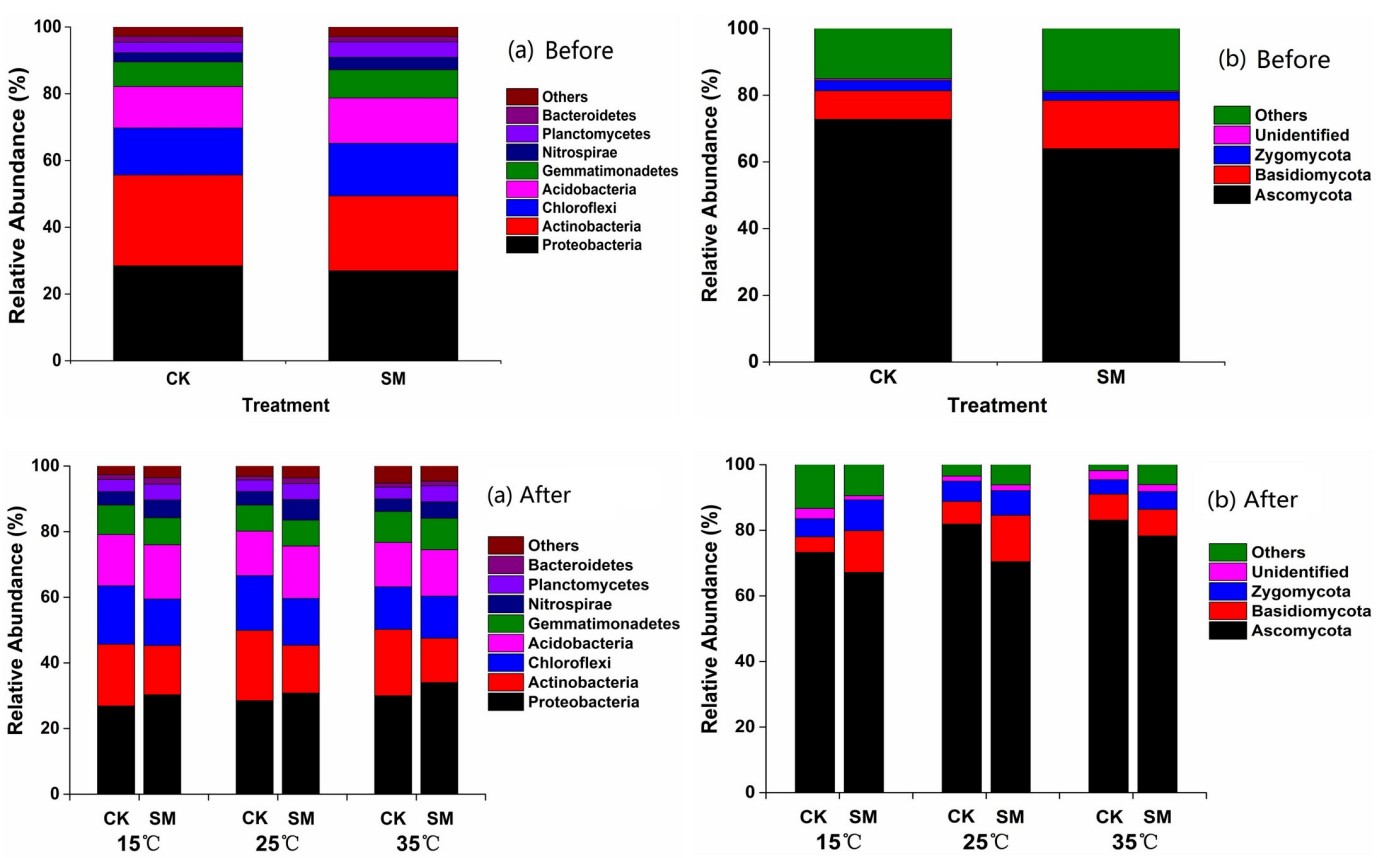

**Fig 6. Relative abundances of (a) bacterial and (b) fungal phyla under straw mulching and no mulching before and after incubation.** CK: no mulching; SM: straw mulching.

respiration increased rapidly at the early stage of incubation (0–15 DAI), decreased rapidly during mid incubation (16–30 DAI), and then decreased relatively slightly during the later stage of incubation (31–40 DAI). These changes in SR suggested that labile organic matter may have depleted quickly along with incubation time as observed in previous studies [6]. After 40 DAI, the respiration rate remained stable due to lower C and N availability. Furthermore, a rapid proliferation of microbial communities might have subsequently allocated more substrates to their proliferation and growth than to respiration, thereby decreasing soil respiration. Thereafter, the soil respiration rate and cumulative $CO_2$-C evolution reached a steady state equilibrium during the day [30].

The daily respiration rate and cumulative respiration were strongly affected by straw mulching and incubation temperatures over a 60-day incubation period in the present study (Figs 1 and 2). Soil respiration was higher with straw mulching than without, similar to results previously reported by Lanza et al. [31]. This higher rate of soil respiration can be explained by a higher quality of organic C in the straw mulch treatment and a higher SOC [5]. Sources of C inputs, including plant litter and rhizodeposition, act as substrates that are mineralized to $CO_2$ by the soil microbial community [32]. Furthermore, there was a clear relationship between respiration rate and SOC and PCM (S6 Table), which was consistent with Lee et al. [19]. Both SOC and PCM played a dominant role in determining the variance in soil respiration, and soil microbial community composition was not the only major determinant of the soil respiration

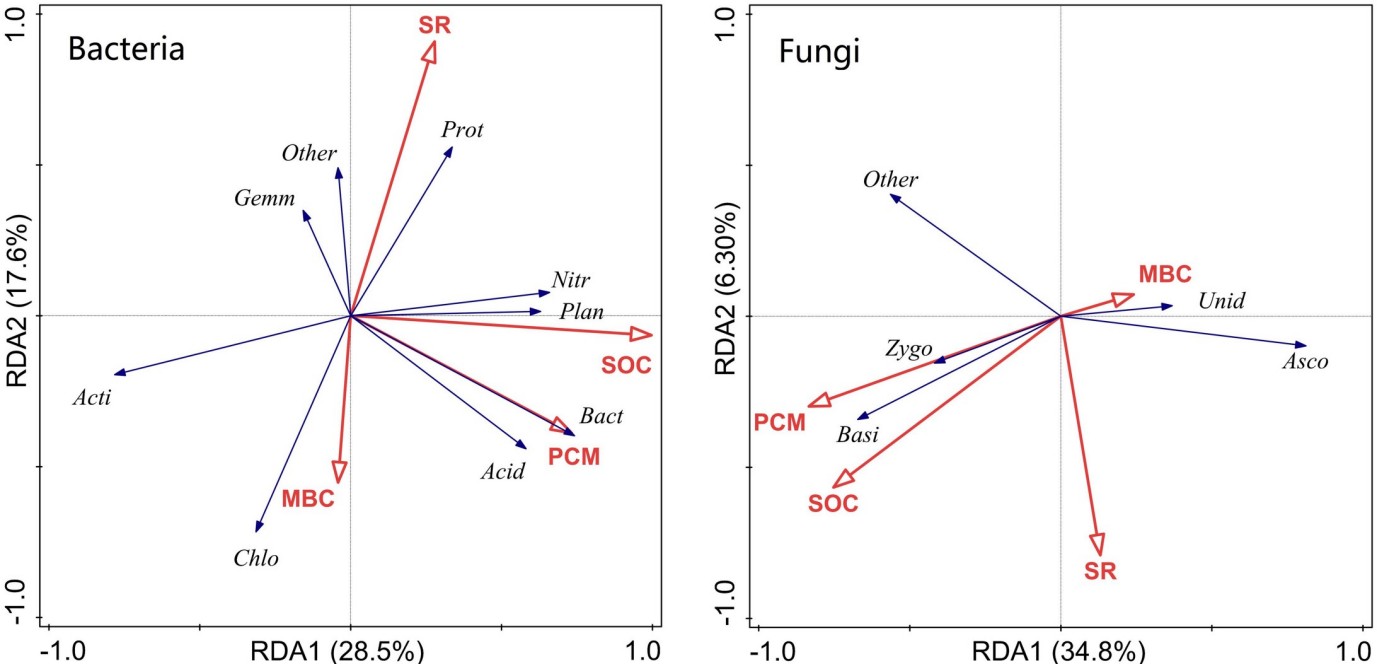

**Fig 7. Plots of the eigenvectors from Redundancy Analysis (RDA) in the plane of the first two axes to show the relations among the microbial populations (black arrows) and soil C fractions and respiration (SR) (red arrows).** Proteobacteria (Prot), Actinobacteria (Acti), Chloroflexi (Chlo), Acidobacteria (Acid), Gemmatimonadetes (Gemm), Nitrospirae (Nitr), Planctomycetes (Plan), Bacteroidetes (Bact), Ascomycota (Asco), Zygomycota (Zygo), Basidiomycota (Basi) and unidentified (unid).

[33]. MBC was not correlated to respiration, which indicated that the microbial respiratory response to carbon additions is not necessarily linked to microbial biomass growth response [34]. However, Józefowska et al. [35] found a negative correlation between C content and respiration, which may be related to the protection of soil organic matter. Soil organic matter may be stabilized in the form of organo-mineral complexes [36], which are resistant to microbial degradation.

Soil respiration increased with increased incubation temperature (Fig 1), which agreed with the results reported by Lee et al. [19] and Suh et al. [37]. However, many warming experiments showed either no change [38] or decreased [39] soil respiration when incubated at higher temperatures. Teklay et al. [16] suggested that once the soil C limitation was alleviated, the effect of temperature became apparent. This difference could also be explained by changes in soil water. Water stress could suppress respiration by decreasing microbial activities directly, but also decrease soil respiration via the inhibition of carbon allocation and substrate availability indirectly [40]. In our experiment, soil water content remained stable, and increasing incubation temperature may reduce the turnover time of labile and recalcitrant C pools and temperature sensitivity [41]. So, a longer-term incubation may be needed to test the effect of temperature on soil respiration.

The $Q_{10}$ values in our study were lower than previously reported by Wang et al. [42], where they found that $Q_{10}$ values ranged from 1.96 to 2.76. This difference was probably due to different soil environments (soil organic matter, moisture, microorganism, and temperature) in the two studies. Meyer et al. [43] also found that forest soils were more sensitive to soil warming than cropland soils. In the last decade, some incubation studies illustrated that soils with high C substrate quality have low $Q_{10}$ [44], which was consistent with our study (Table 3). In our

present study, straw mulching had a lower $Q_{10}$ as compared with no mulching. The higher number of microorganisms in soil with straw mulching may accelerate the consumption of C directly caused by soil respiration. No additional C sources were added during our incubation experiment, causing soil C to become insufficient at the late incubation stage and resulting in the decrease of the $Q_{10}$ under the straw mulching. Similarly, lower $qCO_2$ under straw mulching also confirmed these conclusions (Fig 3). However, in our previous study, we found that the $Q_{10}$ was higher under straw mulching than no mulching, probably due to the driving factors differing between the two studies [45]. Dai et al. [46] also indicated that the temperature sensitivity of organic C mineralization in exogenous C is highly sensitive to SOC, compared with no C input.

Huang et al. [47] reported significant correlations between the bacterial community richness and diversity and soil physicochemical properties. At the beginning of our incubation experiment, the abundance of all microbial taxa did not differ with and without mulching (Table 4). After incubation, soil microbial diversity increased because of increased microbial activity (Tables 1 and 5). However, soil bacterial diversity did not show significant differences between soils with and without straw mulch (except bacterial richness), although the PCA analysis revealed that bacterial and fungi communities distinctly separated from each other for CK and SM at all temperatures. Thus, further soil microbial composition studies are needed for understanding the effect of straw mulching on soil bacteria.

Soil bacteria and fungi were also profoundly different at different incubation temperatures (Fig 5). Lower bacterial and fungal diversities at 35 ˚C than at 15 ˚C and 25 ˚C indicated that the increase in temperature may have temporarily enhanced microbial activities and simultaneously promoted competition, which could eventually result in fewer dominant species at 35 ˚C when labile C was likely exhausted towards the end of the experiment [16]. Similarly, Pettersson & Bååth [48] also found a temperature-dependent changes in soil bacterial community in an 80-day incubation study at 5 ˚C˚C, 20 ˚C, and 30 ˚C. Wu et al. [49] showed that soil microbial biomass, indexed by total phospholipid fatty acid concentration, shifted with temperatures in all soils and decreased with increasing incubation temperature.

In our study, straw mulching markedly changed soil bacterial and fungal community composition (Fig 6). The abundances of phyla *Proteobacteria*, *Acidobacteria*, *Bacteroidetes*, *Nitrospirae*, and *Planctomycetes* increased and the abundance of *Actinobacteria* and *Chloroflexi* decreased in SM relative to CK (Fig 6 and S2 Table), which was partly consistent with the reports by Wang et al. [50]. The phylum *Proteobacteria* is generally enriched in the nutrient-rich conditions and plays a significant role in C and N cycling [51, 52]. In this study, the greater abundance of *Proteobacteria* in soils with straw mulch was probably due to the greater SOC and C availability with the additional C input, which can be energy sources for the growth of this phyla [53]. Similarly, the greater abundance of *Acidobacteria* may be due to more particle C fractions in SM than CK [22]. The variations of *Bacteroidetes* were stimulated by crop roots and are well adapted to labile carbon in soil, thus the greater abundance of *Bacteroidetes* might be due to increases in SOC due to straw mulching. The positive impacts of increased organic matter content on the growth of *Proteobacteria* and *Bacteroidetes* have been previously reported [2, 54]. Also, other specific taxa, especially for phylum *Actinobacteria* and *Chloroflexi*, decreased significantly with SM and drove the negative responses of SOC (Fig 6 and S2 Table), which suggested a possible balance of C dynamics being mediated by these two phyla. It has been well established that classes *Thermomicrobia* and *MB-A2-108*, branches of *Actinobacteria*, were more abundant in CK than SM. Thus, this phylum showed a negative correlation with SOC in our study (Fig 7 and S7 Table), which was consistent with the reports by Ren et al. [2]. As for the fungal community compositions, the abundance of phylum *Ascomycota* was higher under CK than SM, probably because it is involved in soil aggregation [22]. The higher

abundance of phylum *Basidiomycota* in SM was mainly because these taxa are largely saprotrophic and benefit from nutrient enrichment of the soil resulting from the larger amounts of input organic matter associated with straw mulching.

Our correlation analysis between C fractions and DNA abundance of the single taxa yielded a high correlation coefficient for the abundances of phyla *Actinobacteria*, *Acidobacteria*, *Bacteroidetes*, *Nitrospirae*, and *Planctomycetes* (Fig 7 and S7 Table), which confirmed that these taxa play an important role in the degradation of soil organic carbon compounds. This result was partly consistent with Fierer et al. [53] and Lanza et al. [31]. Furthermore, the relationships among soil C fractions, SR, and microbial community compositions were different with and without straw mulch (S7 and S8 Tables), which indicated that temperature could influence the response soil microbial community composition to straw mulching.

## Conclusions

Based on a short-term incubation experiment, our study showed that straw mulching affected soil respiration and microbial community composition at different incubation temperatures. Compared to no mulching, straw mulching significantly increased SOC and PCM concentrations, soil respiration and $q$CO$_2$, decreased the Q$_{10}$ value and changed soil microbial community compositions. Soil respiration increased, while soil C fractions and microbial diversity decreased, with increasing temperature. Such changes depended on the alteration of the bacterial and fungal communities with straw mulching. Furthermore, increasing temperature could change soil C sequestration by changing the relationships among soil respiration, microbial community, and C fractions.

## Supporting information

**S1 Fig. Principal Coordinates Analysis (PCoA) of soil microbial community composition under straw mulching and no mulching before incubation based on Bray-Curtis distances.** CK: no mulching; SM: straw mulching.
(PDF)

**S1 Table. Influence of straw mulching and temperature on soil carbon fractions after short-term incubation.** (a) CK: no mulching; SM: straw mulching. (b) Different lowercase letters indicate significant difference among different mulching methods.
(PDF)

**S2 Table. Influence of straw mulching and temperature on soil bacterial phyla (%) after short-term incubation.** (a) CK: no mulching; SM: straw mulching. (b) Different lowercase letters indicate significant difference among different mulching methods or incubation temperatures. (c) NS, not significant. ***Significant at $P \leq 0.001$; **significant at $P \leq 0.01$; *significant at $P \leq 0.05$.
(PDF)

**S3 Table. Relative abundances (%) of bacterial taxa under straw mulching and no mulching after short-term incubation at different temperatures.** (a) CK: no mulching; SM: straw mulching. (b) Numbers followed by different lowercase letters within a row between straw mulching and no mulching are significantly different at $P = 0.05$ by the least square means test. (c) Numbers followed by different uppercase letters within a row between incubation temperatures are significantly different at $P = 0.05$ by the least square means test.
(PDF)

**S4 Table. Influence of straw mulching and incubation temperature on soil fungal phyla (%) after short-term incubation.** (a) CK: no mulching; SM: straw mulching. (b) Different lowercase letters indicate significant difference among different mulching methods or incubation temperatures. (c) NS, not significant. ***Significant at $P \leq 0.001$; *Significant at $P \leq 0.05$. (PDF)

**S5 Table. Relative abundances (%) of fungal taxa under straw mulching and no mulching after short-term incubation at different temperatures.** (a) CK: no mulching; SM: straw mulching. (b) Numbers followed by different lowercase letters within a row between straw mulching and no mulching are significantly different at $P = 0.05$ by the least square means test. (c) Numbers followed by different uppercase letters within a row at different incubation temperatures are significantly different at $P = 0.05$ by the least square means test. (PDF)

**S6 Table. Relationship between cumulative soil respiration and C fractions at different temperatures.** SOC: soil organic carbon; PCM: potential C mineralization; MBC: microbial biomass carbon. (PDF)

**S7 Table. Spearman's rank correlation coefficients (R) between microbial (i.e., bacterial and fungal) compositions and the soil C fractions and respiration at the phylum level.** (a) CK: no mulching; SM: straw mulching. (b) Soil organic carbon (SOC); dissolved organic carbon (DOC); potential C mineralization (PCM); microbial biomass carbon (MBC); soil respiration (SR). (c) ** Correlation is significant at the 0.01 level; * Correlation is significant at the 0.05 level. (PDF)

**S8 Table. Spearman's rank correlation coefficients (R) between microbial (i.e., bacterial and fungal) composition and the soil C fractions and respiration at the class level.** (a) CK: no mulching; SM: straw mulching. (b) Soil organic carbon (SOC); dissolved organic carbon (DOC); potential C mineralization (PCM); microbial biomass carbon (MBC); soil respiration (SR). (c) ** Correlation is significant at the 0.01 level; * Correlation is significant at the 0.05 level. (PDF)

## Author Contributions

**Data curation:** Xin Fu, Mengyi Xie.

**Formal analysis:** Fazhu Zhao.

**Funding acquisition:** Jun Wang.

**Methodology:** Jun Wang, Fazhu Zhao.

**Resources:** Jun Wang.

**Writing – original draft:** Xin Fu.

**Writing – review & editing:** Jun Wang, Russell Doughty.

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
