## [Decision Letter · Decision Letter 0]

6 Apr 2020

PONE-D-20-04458

Increasing temperature can modify the responses of soil C fractions, soil respiration, and microbial community composition to straw mulching

PLOS ONE

Dear Professor Wang,

Thank you for submitting your manuscript to PLOS ONE. After careful consideration, we feel that it has merit but does not fully meet PLOS ONE’s publication criteria as it currently stands. Therefore, we invite you to submit a revised version of the manuscript that addresses the points raised during the review process.

Please, address all comments from the reviewers. In addition to the articles mentioned by reviewer 1, please also refer to Grazing increases the temperature sensitivity of soil organic matter decomposition in a temperate grassland

in order to improve the introduction.

We would appreciate receiving your revised manuscript by May 21 2020 11:59PM. To enhance the reproducibility of your results, we recommend that if applicable you deposit your laboratory protocols in protocols.io, where a protocol can be assigned its own identifier (DOI) such that it can be cited independently in the future. For instructions see: http://journals.plos.org/plosone/s/submission-guidelines#loc-laboratory-protocols

We look forward to receiving your revised manuscript.

Kind regards,

Jorge Paz-Ferreiro, Ph.D.

Academic Editor

PLOS ONE

Journal Requirements:

2. We noticed you have some minor occurrence of overlapping text with previous publications, which needs to be addressed. In your revision ensure you cite all your sources (including your own works), and quote or rephrase any duplicated text outside the methods section. Further consideration is dependent on these concerns being addressed.

4. We note that you are reporting an analysis of a microarray, next-generation sequencing, or deep sequencing data set. PLOS requires that authors comply with field-specific standards for preparation, recording, and deposition of data in repositories appropriate to their field. Please upload these data to a stable, public repository (such as ArrayExpress, Gene Expression Omnibus (GEO), DNA Data Bank of Japan (DDBJ), NCBI GenBank, NCBI Sequence Read Archive, or EMBL Nucleotide Sequence Database (ENA)). In your revised cover letter, please provide the relevant accession numbers that may be used to access these data. For a full list of recommended repositories, see http://journals.plos.org/plosone/s/data-availability#loc-omics or http://journals.plos.org/plosone/s/data-availability#loc-sequencing.

5. Please provide further details in your Methods section on the sequencing carried out on the samples.

7. Please upload a copy of Figures 7 and 8, to which you refer in your text on page xx. If the figure is no longer to be included as part of the submission please remove all reference to it within the text.

Reviewers' comments:

Reviewer's Responses to Questions

**Comments to the Author**

1. Is the manuscript technically sound, and do the data support the conclusions?

Reviewer #1: Partly

Reviewer #2: Partly

Reviewer #3: Yes

2. Has the statistical analysis been performed appropriately and rigorously? 

Reviewer #1: No

Reviewer #2: Yes

Reviewer #3: Yes

3. Have the authors made all data underlying the findings in their manuscript fully available?

Reviewer #1: No

Reviewer #2: Yes

Reviewer #3: Yes

4. Is the manuscript presented in an intelligible fashion and written in standard English?

Reviewer #1: Yes

Reviewer #2: Yes

Reviewer #3: Yes

5. Review Comments to the Author

Reviewer #1: Increasing temperature can modify the responses of soil C fractions, soil respiration, and microbial community composition to straw mulching

Review

Line 1: How can soil C fraction, soil respiration, and community composition have responses? E The title is misleading. Please restructure the sentence.

Line 19: what mulched?

Line 21-22: Restructure sentence, reads weird.

Line 22: Greater than what?

Line 23: SM was averaged across temperatures?

Line 24: Greater than what?

Line 25: Again, greater than what?

Line31: What is the existing relationship that was modified with increasing temperature?

Line 33: The authors haven’t really looked at carbon pools nor microbial activities, so this is not an appropriate keyword

Line 35: Numbers are available for this, please write the percentages instead of writing a vague word as “substantially”

Line 36: What is meant by small shift? And the sentence structure is wrong.

Line 39-40: “SR is often used to approximate the rate of soil organic carbon (SOC) mineralization and decomposition” Please provide reference. Is this true for all soil orders and latitudes? Avoid making such general statements. I would recommend refer to these and references cited in these papers to develop a deeper understanding on this topic and improve the introduction. https://agupubs.onlinelibrary.wiley.com/doi/pdf/10.1002/2017GB005644, https://www.biogeosciences.net/16/663/2019/, https://www.nature.com/articles/srep18370

Line42-43: This statement makes no sense. If the discussion has rarely been limited, why do this study. Also, which species are the authors alluding to here? Vague sentence.

Line 46: changing how and promoting what soil microbial processes?

Line 48: if previous studies have already studied this, why are the authors doing this experiment? What is unique?

Lines 52-54: dryland carbon dynamics is a new concept introduced here, why?

Line 69: why study soil C fractions? The authors haven’t presented a compelling reason.

Line 72: The authors should provide some background information about mulching practices and why there is a need to study it. What have previous studies reported? Have other studied looked at impact of variable temperature on mulching benefits?

Line 73: What does one mean by changing the relationships?

Line 71-74: What hypotheses are driving these objectives?

Line 134: Where are the codes for sequence analyses? Where have the sequences been deposited? Please make code and data available.

Line 194: OTU composition

Line 224-244 and Figure 5: Are these averages? How were the three replicates handled? It is unclear whether these relative abundances are from one replicate per treatment or average across replicates.

Line 319-320: taxa did not differ due to low microbial activity is a stretch. The environment is not significantly different enough to see a difference.

Line 324: What does one mean by “separated by PCA”?

Line 324-325: I would recommend that the authors refer to studies already conducted to understand the effect of straw mulching on bacteria. https://www.sciencedirect.com/science/article/pii/S092913931931056X#f0030

https://link.springer.com/article/10.1007/s13762-017-1434-8,

https://www.sciencedirect.com/science/article/pii/S1164556318304874

Figure3: How were these ellipses drawn?

Reviewer #2: The authors of “Increasing temperature can modify the responses of soil C fractions, soil respiration, and microbial community composition to straw mulching” present results from an incubation experiment manipulating temperature of soils exposed to straw mulching. Respiration, MBC, and microbial community composition were assessed following short term incubation at three different temperatures. Overall conclusions drawn from results seem appropriate, there are a few comments below about additiona metrics to consider based on data already collected.

Introduction

Line 43 Perhaps “defined” instead of “limited.

The introduction focuses primarily on carbon availability and temperature as strong controls on soil respiration in general. However, the focus on dryland systems is cursory at best (line 53-54). Please include additional details about how this information is important for dryland systems.

Results

Line 167-168: It was stated that cumulative respiration was greater in SM than CK but only 3 values were provided for cumulative respiration at the different temperatures. To which treatment do these three values apply?

Throughout the results there are figure captions listed. (line 172-186, 201-204)

Discussion

The authors discuss changes in respiration at temperature, but there is little reference to the temperature sensitivity calculations they performed. Additionally, was temperature sensitivity calculated at different points throughout the experiment, only 1 value per treatment was included. Additional temperature sensitivity calculations would enable additional comparisons to other studies and provide a standard metric for comparison.

Lines 273-294: The authors describe that it is possible the microorganisms have reduced respiration relative to biomass at lower temperatures, however that calculation does not appear in the manuscript. Respiration per unit biomass (sometimes referred to as Rmass) would be a helpful metric for comparison between temperature and treatments as well as other studies. I think that the authors have MBC data and respiration data for at least some of the time points for this calculation.

Tables and Figures

Figure 1b. Is this graph showing an average soil respiration for CK and SM at each temperature or is this only one of the treatments? If one of the treatments then needs to be labeled as such in the caption. If it is an average of the two treatments I don’t think they can be combined since there was a difference in respiration based on treatment.

Table 2. Between what temperatures was Q10 calculated, 15 to 25, 25 to 35, or 15 to 35?

Reviewer #3: Review of the manuscript PONE-D-20-04458

The manuscript is an interesting study on warming as a drive factor for soil C by using long term straw mulching. It falls in the general scope of the journal and plays a relevant role in further understanding of soil C dynamic through investigates the changes in temperature on soil C fractions, soil respiration, and microbial community.

However, I have certain and major concern about Soil C pools and how the authors relate that to microbial habitat, besides, what was the real decomposition process and how was related to increasing temperature. There is lack of information about straw mulch in terms of composition and C: N Ratio.

Other comments:

-Abbreviations in the text are confusing and need to be more clear

-Heading, subheading, subtitles in the manuscript need to be changed to appropriate format.

-Objectives are not clear, and the second objective is part of the first one. Basically, the study

has more than one objective.

-There is no hypothesis related to the objectives to show the mulching impact and warming

as a drive factor for decomposition and microbial activity.

-Separate table for initial chemical and physical soil properties is needed and it is better

than the way was written in the text.

-Lack of information about mulch application

-Tables need to be revised

-Line 41 – 47: the statement is not clear and needs to be revised

-Line 82: what is STN stands for?

-Line 95-98 need more details about soil sampling,

-Line 103: In what basis you adjust soil sample to 60% water.

-Line 110-115: need citation

-Line 128-132: need to be revised

-Line 207-208 the dominant bacterial phyla total composition doesn’t come out to 100%

-Line 307-308 how and need citation

6. PLOS authors have the option to publish the peer review history of their article (what does this mean?). If published, this will include your full peer review and any attached files.

Reviewer #1: No

Reviewer #2: No

Reviewer #3: Yes: Adel H. Youkhana

---

## [Author Response · Author response to Decision Letter 0]

21 Jun 2020

Responses to Editor

Corrected according to the templates.

2. We noticed you have some minor occurrence of overlapping text with previous publications, which needs to be addressed. In your revision ensure you cite all your sources (including your own works), and quote or rephrase any duplicated text outside the methods section. Further consideration is dependent on these concerns being addressed.

Corrected as suggested. References have been added and relative descriptions have been corrected (Lines 192-206, 234-241 in the Revised Manuscript with Track Changes).

We asked Dr. Russell Doughty, who is working in Division of Geological and Planetary Sciences, California Institute of Technology, Pasadena, CA, USA, for language improvements. Also, he is included in the authors’ list in the revised manuscript.

4. We note that you are reporting an analysis of a microarray, next-generation sequencing, or deep sequencing data set. PLOS requires that authors comply with field-specific standards for preparation, recording, and deposition of data in repositories appropriate to their field. Please upload these data to a stable, public repository (such as ArrayExpress, Gene Expression Omnibus (GEO), DNA Data Bank of Japan (DDBJ), NCBI GenBank, NCBI Sequence Read Archive, or EMBL Nucleotide Sequence Database (ENA)). In your revised cover letter, please provide the relevant accession numbers that may be used to access these data. For a full list of recommended repositories, see http://journals.plos.org/plosone/s/data-availability#loc-omics or http://journals.plos.org/plosone/s/data-availability#loc-sequencing.

The raw sequences for all samples were sent to the Sequence Read Archive (SRA) database of the National Center for Biotechnology Information (NCBI, Bethesda, MD, USA) with the accession numbers SRP260944 for bacteria and SRP261054 for fungi, respectively.

5. Please provide further details in your Methods section on the sequencing carried out on the samples.

Changed. See Lines 242-268.

Done.

7. Please upload a copy of Figures 7 and 8, to which you refer in your text on page xx. If the figure is no longer to be included as part of the submission please remove all reference to it within the text.

Removed. We rechecked this sentence, and the description about Figures 7 and 8 is unnecessary. See Line 407. 

Changed. We have revised Supporting Information according to the guidelines. (Lines 574-623)

Responses to Reviewer #1

Line 1: How can soil C fraction, soil respiration, and community composition have responses? E The title is misleading. Please restructure the sentence.

The title was changed as “Increasing temperature can modify the effect of straw mulching on soil C fractions, soil respiration, and microbial community composition” (Lines 1-2 in the Revised Manuscript with Track Changes).

Line 19: what mulched?

“what” was deleted (Line 22).

Line 21-22: Restructure sentence, reads weird.

The sentence was corrected as “Soil respiration, C fractions and bacterial and fungal community structure were measured.” (Lines 24-25).

Line 22: Greater than what?

The sentence was corrected as “SM had greater soil organic carbon and potential C mineralization and a similar microbial biomass carbon throughout the incubation when compared with CK” (Line 26-28).

Line 23: SM was averaged across temperatures?

Deleted. (Line 28)

Line 24: Greater than what?

Line 25: Again, greater than what?

The sentences have been corrected as “Similar microbial community composition was found in the soils with SM and CK before incubation. However, SM had a greater bacterial richness and the relative abundances of Proteobacteria, Acidobacteria, Nitrospirae, Planctomycetes, Bacteroidetes, and Basidiomycota, but a lower relative abundance of Actinobacteria, Chloroflexi, and Ascomycota than CK after incubation” (Lines 31-35).

Line31: What is the existing relationship that was modified with increasing temperature?

This sentence was corrected as “These findings indicated that the effect of straw mulching on soil C cycling and microbial community structure can be highly modified by increasing temperature.” (Lines 41-43).

Line 33: The authors haven’t really looked at carbon pools nor microbial activities, so this is not an appropriate keyword

The keywords were corrected as “Incubation, mulching, microbes, soil carbon, warming” (Line 44).

Line 35: Numbers are available for this, please write the percentages instead of writing a vague word as “substantially”

This sentence was corrected as “Soil contains around 1500 Pg of organic carbon (C) and plays a major role in the carbon cycling in terrestrial ecosystems” (Lines 47-48).

Line 36: What is meant by small shift? And the sentence structure is wrong.

This sentence was corrected as “A small variation in soil C sequestration can lead to a significant change in atmospheric CO2 concentration” (Lines 49-50).

Line 39-40: “SR is often used to approximate the rate of soil organic carbon (SOC) mineralization and decomposition” Please provide reference. Is this true for all soil orders and latitudes? Avoid making such general statements. I would recommend refer to these and references cited in these papers to develop a deeper understanding on this topic and improve the introduction. https://agupubs.onlinelibrary.wiley.com/doi/pdf/10.1002/2017GB005644, https://www.biogeosciences.net/16/663/2019/, https://www.nature.com/articles/srep18370

(1) The sentence was deleted and reviewed. (Line 86-87)

(2) The papers of Wei et al. (2015) and Meyer et al. (2018) were cited in the manuscript. 

Line42-43: This statement makes no sense. If the discussion has rarely been limited, why do this study. Also, which species are the authors alluding to here? Vague sentence.

The sentence was deleted.

Line 46: changing how and promoting what soil microbial processes?

Deleted.

Line 48: if previous studies have already studied this, why are the authors doing this experiment? What is unique?

Though previous studies have already studied the responses of SOC, SR and microbial activity to warming have gained more attentions recently. The comprehensive responses of C soil fractions, SR and microbial communities to straw mulching would vary with different temperature change is now well reported. (Lines 71-73)

Lines 52-54: dryland carbon dynamics is a new concept introduced here, why?

we have deleted the concept of “dryland carbon dynamics”; the straw mulching was widely application in the dryland, so we used the concept, but the concept of “dryland carbon dynamics” in manuscript was not inaccurate. 

Line 69: why study soil C fractions? The authors haven’t presented a compelling reason.

We have added the reason why study soil C fractions as “Generally, SOC would change slowly with management practices due to its large pool sizes and inherent spatial variability. Soil labile C fractions, such as microbial biomass carbon (MBC) and potential C mineralization (PCM) would response more rapidly to environmental change than SOC.” (Lines 56-59).

Line 72: The authors should provide some background information about mulching practices and why there is a need to study it. What have previous studies reported? Have other studied looked at impact of variable temperature on mulching benefits?

Reviewed. The background information about mulching practices and the previous results about temperature on mulching have added as “In the recent decades, straw mulching has widely been adopted to conserve soil water, regulate soil temperature and increase crop yield in dryland cropping systems. The application of straw mulch also has been proposed as a method to store organic carbon long term. In an 8-yr study in the Loess Plateau of China, Wang et al. reported that soil organic C (SOC) stock was 7-35% greater with straw mulching than without”. (Lines 51-56)

Line 73: What does one mean by changing the relationships?

Changed. We have revised the sentence from “explore whether temperature affected soil C sequestration by changing the relationships among soil respiration, microorganisms, and C fractions” to “In the context of climate change, the responses of SOC, SR and microbial activity to warming have gained more attentions recently. Increasing temperature would stimulate soil microbial metabolisms, accelerate SOC decomposition and increase C efflux through SR. Since the temperature sensitivity of SR (Q10) varied with the substrate availability, how the comprehensive responses of C soil fractions, SR and microbial communities to straw mulching would vary with different temperature change is now well reported.”. (Lines 71-76)

Line 71-74: What hypotheses are driving these objectives?

We have added two hypotheses as “(1) straw mulching would increase SR rates by increasing the input of organic matter in the soil compared with no mulching; and (2) straw mulching could change the temperature sensitivity of the agro-ecosystem due to the regulation of soil C fractions and soil microbial community.”. (Lines 80-83).

Line 134: Where are the codes for sequence analyses? Where have the sequences been deposited? Please make code and data available.

Added. The raw sequences for all samples have sent to the Sequence Read Archive (SRA) database of the National Center for Biotechnology Information (NCBI, Bethesda, MD, USA) under the accession numbers SRP260944 and SRP261054 for bacteria and fungi, respectively. (Line 268)

Line 194: OTU composition

Changed. “OUT” was changed to “OTU”. (Line 365).

Line 224-244 and Figure 5: Are these averages? How were the three replicates handled? It is unclear whether these relative abundances are from one replicate per treatment or average across replicates.

Figure 5 data showed the average across three replicates.

Line 319-320: taxa did not differ due to low microbial activity is a stretch. The environment is not significantly different enough to see a difference

The sentence “which is probably due to lower microbial activity” was removed. (Line 516).

Line 324: What does one mean by “separated by PCA”?

We have revised the sentence as “According to PCA analysis, bacterial and fungal communities were significantly different from each other with and without straw mulch and for the incubation temperatures”. (Lines 519-520).

Line 324-325: I would recommend that the authors refer to studies already conducted to understand the effect of straw mulching on bacteria. https://www.sciencedirect.com/science/article/pii/S092913931931056X#f0030

https://link.springer.com/article/10.1007/s13762-017-1434-8,

https://www.sciencedirect.com/science/article/pii/S1164556318304874

Added. We have added the references of Fu et al. (2019); Huang et al. (2019); Wang et al. (2020).

Figure3: How were these ellipses drawn?

These ellipses represent significant differences between different treatments. The purpose of these ellipses is making readers see the difference of two treatments more clearly. The method was refered to Hemkemeyer et al. (2015). 

Responses to Reviewer #2

Introduction

Line 43 Perhaps “defined” instead of “limited”.

Agree. We have deleted the sentence. (Line 123 in the Revised Manuscript with Track Changes)

The introduction focuses primarily on carbon availability and temperature as strong controls on soil respiration in general. However, the focus on dryland systems is cursory at best (line 53-54). Please include additional details about how this information is important for dryland systems.

We have added the description about the development of straw mulching in dryland. (Lines 51-56)

Results

Line 167-168: It was stated that cumulative respiration was greater in SM than CK but only 3 values were provided for cumulative respiration at the different temperatures. To which treatment do these three values apply? Throughout the results there are figure captions listed. (line 172-186, 201-204)

Each value is an average of three replicates for every treatment. We have added figs captions. (Lines 239, 243-245, 253, 284, 288 and 306)

Discussion

The authors discuss changes in respiration at temperature, but there is little reference to the temperature sensitivity calculations they performed. Additionally, was temperature sensitivity calculated at different points throughout the experiment, only 1 value per treatment was included. Additional temperature sensitivity calculations would enable additional comparisons to other studies and provide a standard metric for comparison.

We have added the discussion about temperature sensitivity and compared with other studies as “ The result by Wang et al. showed that Q10 values in all soils ranged from 1.96 to 2.76, which was higher than our study. This was probably due to different soil environments (soil organic matter, moisture, microorganism, temperature) compared with our study. In the recent decade, some incubation studies illustrated that soils with high C substrate quality have low Q10, which was consistent with our study (Table 3). In our present study, straw mulching had lower Q10 as compared with no mulching. Higher soil microorganisms under straw mulching may accelerate the consumption of C directly caused by soil respiration. But no additional C sources were added during the incubation experiment, causing soil C is insufficient at the late incubation stage, and resulting in the decrease of the Q10 under the straw mulching. Similarly, lower qCO2 under straw mulching also confirmed these conclusions (Fig 3). However, in our previous study, we found that the Q10 was higher under straw mulching than no mulching, probably due to the driver factors was different as the present study. Dai et al. also indicated that the temperature sensitivity of organic C mineralization in the exogenous C is highly sensitive to SOC, compared with no C input.”. (Lines 495-512)

Lines 273-294: The authors describe that it is possible the microorganisms have reduced respiration relative to biomass at lower temperatures, however that calculation does not appear in the manuscript. Respiration per unit biomass (sometimes referred to as Rmass) would be a helpful metric for comparison between temperature and treatments as well as other studies. I think that the authors have MBC data and respiration data for at least some of the time points for this calculation.

We have added the description about metabolic quotient (qCO2), and found “ The qCO2 under both CK and SM appeared to increase with increasing incubation temperatures (Fig 3). For all incubation temperatures, SM had higher qCO2 than CK.”. (Lines 322-324 and Fig 3.)

Tables and Figures

Figure 1b. Is this graph showing an average soil respiration for CK and SM at each temperature or is this only one of the treatments? If one of the treatments then needs to be labeled as such in the caption. If it is an average of the two treatments I don’t think they can be combined since there was a difference in respiration based on treatment.

It is an average of the two treatments, yes, there was a difference in respiration based on different treatments. We are agreed with this view and have revised the figure, which showed Line 326

Table 2. Between what temperatures was Q10 calculated, 15 to 25, 25 to 35, or 15 to 35?

Q10 was calculated from 15 to 25, 25 to 35. We have added the formulas and reference. (Lines 282-289)

Responses to Reviewer #3

However, I have certain and major concern about Soil C pools and how the authors relate that to microbial habitat, besides, what was the real decomposition process and how was related to increasing temperature. There is lack of information about straw mulch in terms of composition and C: N Ratio.

Reviewed (1) The manuscript mainly focuses on the response of carbon pools under straw mulching to global warming. Therefore, we mainly analyzed the influence of straw mulching on soil C fractions, SR rate and soil microbial community. 

(2) The information about straw mulch in term of composition and C/R ratio was added as “Corn straw had a C/N ratio of 40.1 and the contents of cellulose, hemicellulose and lignin were 32%, 28% and 15.5%, respectively.”. (Lines 197-198).

Other comments:

-Abbreviations in the text are confusing and need to be more clear

Reviewed.

-Heading, subheading, subtitles in the manuscript need to be changed to appropriate format.

Changed.

-Objectives are not clear, and the second objective is part of the first one. Basically, the study has more than one objective.

Objectives were revised as “(1) determine changes in carbon fractions and SR rates to different temperatures in soils with and without straw mulch, (2) quantify the effect of straw mulch on soil temperature sensitivity; (3) explore the relationships among soil C fractions, SR rates and soil microbial community in a wheat cropping system in the Loess Plateau, China” (Lines 90-94).

-There is no hypothesis related to the objectives to show the mulching impact and warming as a drive factor for decomposition and microbial activity.

Two hypothesis were added as “(1) determine changes in carbon fractions and SR rates to different temperatures in soils with and without 9-yr straw mulching, (2) quantify the effect of straw mulching on soil temperature sensitivity; and (3) explore the relationships among soil C fractions, SR rates and the soil microbial community.”. (Lines 77-80).

-Separate table for initial chemical and physical soil properties is needed and it is better than the way was written in the text.

The tables about soil C fractions was revised. We have separated the data of initial chemical and physical soil properties from after incubation. See Tables 1 and 4.

-Lack of information about mulch application

The information about mulch application was added as “In the recent decades, straw mulching has widely been adopted to conserve soil water, regulate soil temperature and increase crop yield in dryland cropping systems. The application of straw mulch also has been proposed as a method to store organic carbon long term.” (Lines 51-54).

-Tables need to be revised

Reviewed. See Tables 1, 2, 4 and 5.

-Line 41 – 47: the statement is not clear and needs to be revised

The descriptions about the development of temperature on SR and microbial community was revised as “A comprehensive understanding of how soil respiration is affected by changing different factors is important to improving soil carbon sequestration. However, positive, neutral, and negative effects of temperature on soil C fractions and SR have been documented. Thus the temperature dependency of soil C decomposition and its possible future variation needs to be more clearly defined.” and “In the context of climate change, the responses of SOC, SR and microbial activity to warming have gained more attentions recently. Increasing temperature would stimulate soil microbial metabolisms, accelerate SOC decomposition and increase C efflux through SR.”. (Lines 71-73)

-Line 82: what is STN stands for?

We have changed “STN” to “soil total nitrogen content”. (Line 190)

-Line 95-98 need more details about soil sampling,

The details about soil sampling was added as “Fresh soil samples were collected from after corn harvest in October 2017. Soil samples (about 10 kg) were collected with a spade from the surface layer (0-20 cm) from five places within a plot. Then we composited five samples to one sample, and placed them in plastic boxes. We tried to avoid damaging the soil structure during the collection process.”. (Lines 207-211)

-Line 103: In what basis you adjust soil sample to 60% water.

The cumulative soil respiration is maximum when the water content reached 60% of the field water holding capacity. Therefore, during the incubation period, we used weighing method to keep the soil water content. The citation was added in Line 218.

-Line 110-115: need citation

The citation was added. See Lines 221.

-Line 128-132: need to be revised

The methods section on the sequencing was revised as “Soil DNA was extracted from 0.5 g of freeze-dried soil using Fast DNA SPIN extraction kits (MP Biomedicals, Santa Ana, CA, USA). The extraction method was same as Ren et al. The universal Eubacterial primers 338F (5’-ACTCCTACGGGAGGCAGCA) and 806R (5’-GGACTACHVGGGTWTCTAAT-3’) were used for amplifying the 16S rRNA V3-V4 fragment. The universal eukaryotic primers ITS5F (5’-GGAAGTAAAAGTCGTAACAAGG) and ITS1R (GCTGCGTTCTTCATCGATGC) were used for amplifying the ITS-1 region.”. (Lines 154-159)

-Line 207-208 the dominant bacterial phyla total composition doesn’t come out to 100%

The dominant bacterial phyla total composition count for about 95%, and the other bacterial phyla count for about 5%. See Line 378.

-Line 307-308 how and need citation

 The need citation was added as “Soil organic matter may be stabilized in the form of organo-mineral complexes, which are resistant to microbial degradation.”. (Lines 482-483)

---

## [Decision Letter · Decision Letter 1]

23 Jul 2020

Increasing temperature can modify the effect of straw mulching on soil C fractions, soil respiration, and microbial community composition

PONE-D-20-04458R1

Dear Dr. Wang,

We’re pleased to inform you that your manuscript has been judged scientifically suitable for publication and will be formally accepted for publication once it meets all outstanding technical requirements.

Kind regards,

Jorge Paz-Ferreiro, Ph.D.

Academic Editor

PLOS ONE

Additional Editor Comments (optional):

Reviewers' comments:

Reviewer's Responses to Questions

**Comments to the Author**

1. If the authors have adequately addressed your comments raised in a previous round of review and you feel that this manuscript is now acceptable for publication, you may indicate that here to bypass the “Comments to the Author” section, enter your conflict of interest statement in the “Confidential to Editor” section, and submit your "Accept" recommendation.

Reviewer #1: All comments have been addressed

Reviewer #3: All comments have been addressed

2. Is the manuscript technically sound, and do the data support the conclusions?

Reviewer #1: Yes

Reviewer #3: Yes

3. Has the statistical analysis been performed appropriately and rigorously? 

Reviewer #1: Yes

Reviewer #3: Yes

4. Have the authors made all data underlying the findings in their manuscript fully available?

Reviewer #1: Yes

Reviewer #3: Yes

5. Is the manuscript presented in an intelligible fashion and written in standard English?

Reviewer #1: Yes

Reviewer #3: Yes

6. Review Comments to the Author

Reviewer #1: I agree with the responses to the review document and the corrections made in the manuscript. The quality is much improved now.

Reviewer #3: (No Response)

7. PLOS authors have the option to publish the peer review history of their article (what does this mean?). If published, this will include your full peer review and any attached files.

Reviewer #1: No

Reviewer #3: **Yes: **ADEL H YOUKHANA

---

## [Editor Report · Acceptance letter]

30 Jul 2020

PONE-D-20-04458R1 

Increasing temperature can modify the effect of straw mulching on soil C fractions, soil respiration, and microbial community composition 

Dear Dr. Wang:

I'm pleased to inform you that your manuscript has been deemed suitable for publication in PLOS ONE. Congratulations! Your manuscript is now with our production department. 

Kind regards, 

on behalf of

Dr. Jorge Paz-Ferreiro 

Academic Editor

PLOS ONE